# The Multi-Pattern Approach for Systematic Analysis of Transition Pathways

**Fjalar J. de Haan** [1,*,†] and **Briony C. Rogers** [2,†]

1    Melbourne School of Design, Faculty of Architecture, Building and Planning, The University of Melbourne, Parkville, VIC 3010, Australia
2    School of Social Sciences, Faculty of Arts, Monash University, Clayton, VIC 3800, Australia; briony.rogers@monash.edu
*    Correspondence: fjalar@fjalar.org
†    These authors contributed equally to this work.

**Abstract:** Pathways have become a central notion in various areas of research, amongst which are the studies of transitions to sustainability. Though various typologies and concepts are available, a framework for systematic analysis of transition pathways is lacking. We present the Multi-Pattern Approach (MPA) to fill this lacuna and provide a step-by-step manual for its application. The MPA addresses a range of traditional challenges of transitions' pathway analysis, such as temporal and functional system demarcation and the unravelling of complex, interrelated systemic storylines. The approach provides an oft-called for rigour which allows a diagrammatic and formulaic representation of transitions' pathways. Because of these qualities, the approach allows systematic cross-case comparison and provides a bridge between narrative-based and computational transitions research. The approach is demonstrated with an in-depth empirical case study of water management in Melbourne, Australia over the last 180 years. The article first presents a high-level mapping of the system's evolution over time and a detailed analysis of the uptake and phasing out of specific servicing technologies and practices.

**Keywords:** sustainability transitions; pathways; multi-pattern approach; urban water; analysis; modelling

---

## 1. Introduction

Pathways have become a central and perhaps unifying concept in several areas of research into transformation and long-term trajectories of change. Rosenbloom [1] speaks of its promise as a bridging concept' in the context of low-carbon transitions and Wise et al. [2] observe that it has gained traction in a variety of discourses and policy domains'. In the field of sustainability transitions, pathways have long been a crucial notion. It can be found in the transitions literature as far back and foundational as René Kemp's [3] article on regime shifts and in Jan Rotmans and colleagues' [4] *More Evolution than Revolution* article—moreover, references in the former article suggest that the use of terminology like 'pathways' or 'paths' was already well established. Pathways then referred to the overall temporal unfolding of transitions and this intuitive meaning still applies to its present usage. Focussing on pathways means taking a transition-at-large one's unit of analysis. That this is an appealing approach is apparent from the volume of publications it generates including such relevant recent examples as [5–8]. In the sub-field of transitions modelling (see Holtz et al.'s [9] appraisal), one could identify a class of 'pathway models' (a term due to Holtz, we think) of which the Matisse model [10,11] and the STM [12] are key examples.

The *analysis* of transition pathways, then, seems an important and worthwhile endeavour. Various typologies have been developed that could be a starting point for pathway analysis, of which Geels and Schot's [13] is probably the best known. It is also exceptional in the sense that it distinguishes its

types according to the various dynamics that make up a pathway, rather than assessing the overall 'shape' (like van der Brugge and Rotmans [14] do) or using whole-of-transition parameters such as 'coordination' (as Smith et al. [15] and Rotmans [16] do). Typologies are a useful, if coarse, instrument for pathway analysis—one of the obvious limitations being that an entire transition needs to fit a certain type, which is a big ask. Geels et al. [8] confront this issue in two case studies by proposing that transitions may 'shift' between pathways, a possibility which already had a prelude in the Geels and Schot [13] article. This at the same time acknowledges the problem *and* devaluates the typology—if a transition cannot meaningfully be classified as one of the types in a typology, what is its value? The idea of pathways being composites or sequences however is very useful but it does require a smaller unit of analysis—a sort of sub-pathway as it were.

This article presents a theoretical framework for *systematic analysis* of transition pathways: the Multi-Pattern Approach (MPA). The MPA separates, and then combines, the analysis of system *state* and system *change*. Analysis of system state entails breaking down the system with respect to the different solutions provided for different societal needs. This yields a system composition in terms of what will be called *constellations*. Analysis of system change entails breaking down the transition pathway into a sequence, or sequences, of *patterns*—ideal-typical units of system change, as it were. Hence Multi-Pattern Approach. In this *modular* fashion, the MPA enables systematic mapping of complex transitions story lines—be they based on qualitative or mixed data—without sacrificing complexity for overview.

The MPA was first introduced by de Haan [17] and de Haan and Rotmans [18], with origins going further back to a 2007 conference contribution [19]. It has since been applied to cases as varied as health care, urban water management and energy. A key aim of the approach was to provide a bridge between narrative and modelling research on transitions, hence there are several computational and mathematical applications of the MPA—in full and in part—like in [12,20,21]. These experiences, in particular the collaborations between the authors of this article, have produced several refinements, increased consistency and enabled some simplifications—at least in our view. We therefore thought it opportune to present an updated MPA to consolidate the improvements and provide a point of departure for future work. The emphasis in this article shall be on the MPA as a *methodology*, in slight contrast to earlier expositions focussing more on the theoretical aspects. Where appropriate, deviations in terminology and interpretation from previous versions will be discussed.

Some of the extant applications illustrate how the MPA, by design, addresses—or rather, avoids or circumvents—a number of challenges faced by transition analysts:

By analysing system change as sequences of patterns, the MPA avoids the issue of arbitrary temporal delineations (when did the transition begin, when did it end?). Indeed, some applications have exploited the possibility to 'zoom in' on parts of (ongoing) transitions without the need to consider the beginning or end of the larger process, e.g., for monitoring of transition progress [22,23] or for assessing the plausible changes ahead [24].

The MPA has a central analytical role for the societal needs the various constellations are meant to meet, thus avoiding a technology focus. Indeed, the first applications of the MPA were on cases of transitions in health care systems (see e.g., [25,26] which report on research carried out in 2007–2008). Transition pathways are not restricted to innovation journeys alone and the MPA therefore explicitly frames top-down, imposed transformation as a possible pattern of change—a recurring pattern in the health care transitions referred to, and a conceptual requirement to appropriately analyse large-scale policy interventions.

By analysing the system state with the aid of constellations, the MPA avoids having to introduce 'levels' in cases where they do not apply or obscure matters. Constellations can function alongside each other without one being an innovation supposed to overtake the status quo embodied by the other. In mobility, one can think of public and car-based transport coexisting and evolving, and in water management the constellations around sanitation, drainage, supply, decentralised green infrastructures

all concurrently meet water-related needs (see [27], for a water case). Also in the health care analyses referred to before, several constellations (*ten* in [26]) compose the bigger picture.

Though the MPA is a framework for analysis, it is useful in a policy making-, practice- or action-oriented setting. The MPA, or elements of it at least, have been used in Transition Management processes—for example, in the earlier mentioned monitoring efforts but also in the participative arena process (see Loorbach [28] for a comprehensive overview of Transition Management processes and concepts). Extensive reflection on how the perceived dynamics and the pattern(s) desired to pursue can shape the Transition Management processes is offered by ([25], pp. 243, 246). Other ways in which the MPA would be useful in Transition Management processes (though we do not know if, or how extensively, it has been employed in these ways) are (1) to frame the shared systems perspective and structure the problem in transition arenas, and (2) to develop transition pathways, in these contexts serving as 'inspiring storylines that include goals and interventions on the short, mid and long term' ([29], pp. 46–47).

To do right by its name, the MPA will be presented *as an approach*. In fact, this article should be able to function as a manual for its application to an analyst's case of choice. Once the 'manual' has been introduced, the article provides two in-depth empirical applications of the MPA to illustrate the approach. The two applications are on the same case—transitions in urban water management in Melbourne, Australia—but with different levels of scope and detail—one broad and encompassing, one narrower at a higher resolution. It is worth stressing that the empirical application is not meant to 'validate' the MPA as a theory. The MPA is a framework for transition pathway analysis and the cases provide illustrations of how to apply it and the kind of insights that can be gained with it.

## 2. MPA as a Framework for Systematic Analysis of Transitions

### 2.1. Core Concepts—System State

Before the MPA can be presented as a stepwise approach, its core concepts and their relations need to be defined. The starting point for this should probably be what the likely starting point of any pathway analysis would be the *system* under study. Figure 1 presents a diagrammatic representation of system state using various concepts about to be introduced.

### 2.1.1. Systems and Societal Needs

By *system*, it is meant, in the MPA, a service- and, or resource-provision system. These are the systems designed and evolved to meet human needs on societal scales—or *societal needs*, for short [30]. Thus, there are mobility systems meeting mobility needs, education systems meeting education needs, agro-food systems meeting the needs related to nutrition and so on. This notion of system is broader than that of a socio-technical system, as for instance used by Geels and Schot [13], which is more narrowly defined around some key technology. It is however identical to that of ([16], see p. 59 for examples). In many cases, there is a hierarchy of needs related to what could be called the primary need the system is intended to meet. For example, mobility systems do not only serve to meet the basic need for transport of people and goods but also higher order needs such as social connectivity (see [30], for more on this).

### 2.1.2. Solutions and Institutions

To meet societal needs, various *solutions* exist and new ones continue to be invented. Some of these involve tools (e.g., technologies and infrastructures), some of these involve processes (e.g., somewhat routinised or standardised practices) or combinations of these (see [31–33]). Examples range from tool-based solutions such as the provision of reticulated dam water (in water management) and trams as a means of public transport (mobility systems), to process-based solutions like general practitioner house calls and the 'talking therapy' approach (general and mental health care respectively) and lecture-based knowledge transfer (education). To make such solutions work systematically, either

on their own or in combination and interaction, they need to be accompanied by *institutions*—that is, organisations, norms, rules and regulations that enable them to perform their need-meeting functions. Note that early MPA publications, like [18], did not use the solutions and institutions conceptualisation. This framing was developed by the authors in the context of their joint research that involved both systematic narrative development and modelling. See e.g., [12,31].

### 2.1.3. Constellations

Not all solutions work the same and not all work together well. Often the organisations and other institutions required to make one solution work are not suitable for other solutions. This can be for technical reasons (a nuclear power plant and rooftop solar panels both generate electricity but suit different organisational forms), for institutional reasons (public service models precluding some private options and vice versa, e.g., in mobility, or safety and quality standards allowing only certain certified solutions as in health care) and other reasons like ideological incompatibilities, traditions and organisational cultures and so on. The consequence of this is that solutions seem to come as 'package deals'. Some solutions are managed and regulated well and easily together and fit a similar service model, which allows for economies of scale and organisational efficiencies—think of buses, trams and trains sharing ticketing systems and having synchronised timetables and think of various distinct medical specialties accessed similarly by a patient (GP referral) and financed by similar means (e.g., universal health care financing systems like NHS in the UK and Medicare in Australia). Such package deals of solutions are called *constellations*.

Some constellations meet a larger share of societal needs than others and this can be said to be a measure of its *power* [18]. Change in the power of constellations over time is of course a key concern of any MPA analysis. Power is a multifarious and contested concept and this conception will surely not meet universal approval, so some notes seem appropriate. For other, not necessarily conflicting, conceptualisations of power in the context of transitions, see [34–37]. In the MPA conceptualisation, the power of a constellation depends both on the amount of technical and organisational capacity it has in place, *as well as on how much it is needed*. If an entire society would switch swiftly away from one constellation's solution set (say, fossily fueled electricity) to an alternative entirely, that constellation is rendered powerless. Whether this allows some kind of zero-sum interpretation is not a simple matter to say. If need-meeting is equated with a market share, perhaps. (See de Haan [20] for an explicitly zero-sum interpretation of constellation power and a mathematical treatment of particular consequences.) If more abstract needs, like beauty or equity, are involved, an obvious measure of need-meeting capacity seems unavailable. At any rate, this conceptualisation does lend itself—at least in principle—to a quantitative treatment, though again the more abstract needs may be challenging. A quantitative implementation as part of a simulation model was presented by de Haan et al. [12].

Clearly, a constellation is a service-provision system in its own right, albeit one with what one could call institutional coherence. The constellation is an important unit of analysis in the MPA, though not the only one—more on this later. Examples of constellations are «public transport» in mobility systems, «public sanitation» in water management, «dental health» in health care and «the university» in education. The «constellation name» notation using guillemets (the « ») was introduced by van Raak and de Haan [26] to clearly mark that a constellation was referred to and not the profession, practice, solution or technology it could possibly be confused with. The use of the guillemets is intended to be somewhat similar to using quotation marks to signal that a word is being used *as a word*, instead of as its meaning (like in: the word 'word' is a noun). Thus, we can say: 'in certain systems «public transport» is a constellation'. This, in contrast to: 'there are several modes of public transport'.

Though all constellations are service-provision systems, not all service-provision systems are single constellations—many are composites, consisting of several constellations. A key feature of the MPA is the ability to disaggregate societal systems in their composing constellations. This also allows the analyst to make an informed decision about focussing on only those few constellations that are the protagonists in the transition under study, with the 'package deal' criterion avoiding ambiguities

about what is in and what is not. For example, van Raak and de Haan [26] could justify leaving «dental health» and «complementary medicine» out of the analysis as they only played a minor role in the overall story—their no doubt interesting and intricate histories notwithstanding.

Note that it is not at all necessary for a service-provision system to consist of one dominant constellation and one or several constellations that are contestants to its position. The Multi-Level Perspective (as per Geels [38] for example) terminology of regimes and niches can readily be used in MPA applications, with niches corresponding to constellations with (relatively) little power and the regime as the constellation, or group of constellations, having greatest power. There is, however, no necessity for those 'niches' to be new or innovative, nor do they need be potential usurpers of the regime throne. In many mobility systems, «the car» constellation is arguably the regime, but «public transport» often pre-dates its ascendance and can hardly be deemed an innovation any more (though there may certainly be public transport innovations), or, in Dutch health care, the «mental health» constellation never seriously competed with the «medical specialties» though both innovated and gained power (see e.g., [26]).

### 2.1.4. Conditions

Though not a proper part of the description of the system state, the circumstances in which the system functions are an obviously relevant aspect of a transitions analysis because amongst those circumstances could be its putative causes. Without an actual *theory* of transformative change, however, the issue is that so much of the environment of a system might be relevant as to make it impractical to consider. The MPA as presented here tries to refrain from imposing theoretical constraints—such as which environmental factors may bring about transformative change—as much as reasonably possible. A different approach to this issue is to gather all such environmental factors under one header and infer ad hoc from the case narrative which should be included as causes, as is done in the MLP with 'landscape pressures'. This, however, precludes, or at least biases, post hoc theorising as to what *types* of landscape pressures are correlated to manifestations of transformative change. See Martínez Arranz [39] for a meta-analysis of empirical cases investigating the various landscape pressures in transitions.

The middle way the MPA takes here is to be guided by the societal *needs* met by the various constellations and the environmental *constraints* of their functioning. The evolution of constellations then is considered a process of adaptation against the changing environment of societal needs and constraints. Note that this is *not* excluding factors like political whim or ideology as potential drivers of transitions and it is also certainly not precluding the possibility that change may be for the worse, i.e., maladaptation. The MPA describes *patterns* of change but remains agnostic as to how societal actors accomplish this change or what their motivation is. The idea is that the MPA provides the analytical tools to unravel transition pathways into small enough patterns so the analyst is better equipped to answer precisely such questions as 'who did what and why?' and for the theoretician to recognise generalities across cases.

In previous publications on the MPA, notably de Haan and Rotmans [18], the conditions for transformative change were theorised one step further into three main categories. The categories were: *tension*—adverse functioning with respect to the environment, *stress*—compromised functioning within the constellation, and *pressure*—from competition with another constellation. The reasoning regarding actions and motivations of actors applies unchanged (p. 92).

Here, we describe the conditions under which transformative change occurs in terms of societal *needs* and *constraints*.

**Societal needs** We discussed societal needs in Section 2.1.1 as they are an important guide in the analysis of the system itself. Societal needs become conditions for transformative change if they are either *under*met (or not met at all) or *over*met. In the former, the societal system is expected to adapt to provide a solution that meets the need, or at least more of it, whereas, in the latter case, one or more solutions can be expected to be phased out).

**Constraints** By constraints, we mean conditions that hinder, limit or otherwise constrain the
functioning of certain solutions. Straightforward examples are when a solution is costly (relative
to alternatives) or dependent on another scarce resource. Note that it depends on the specific
contextual conditions whether this actually constitutes a constraint—a solution being costly may
not be problematic at all in a sufficiently affluent society. Other examples of constraints are
those involving public disapproval (e.g., solutions involving nuclear energy or child-labour),
organisational complexity (certain solutions may be hard to manage) or solutions not scaling
with growing or reduced demand. Here also, it will depend on context as to what extent they
constitute actual constraints to the functioning of the solutions.

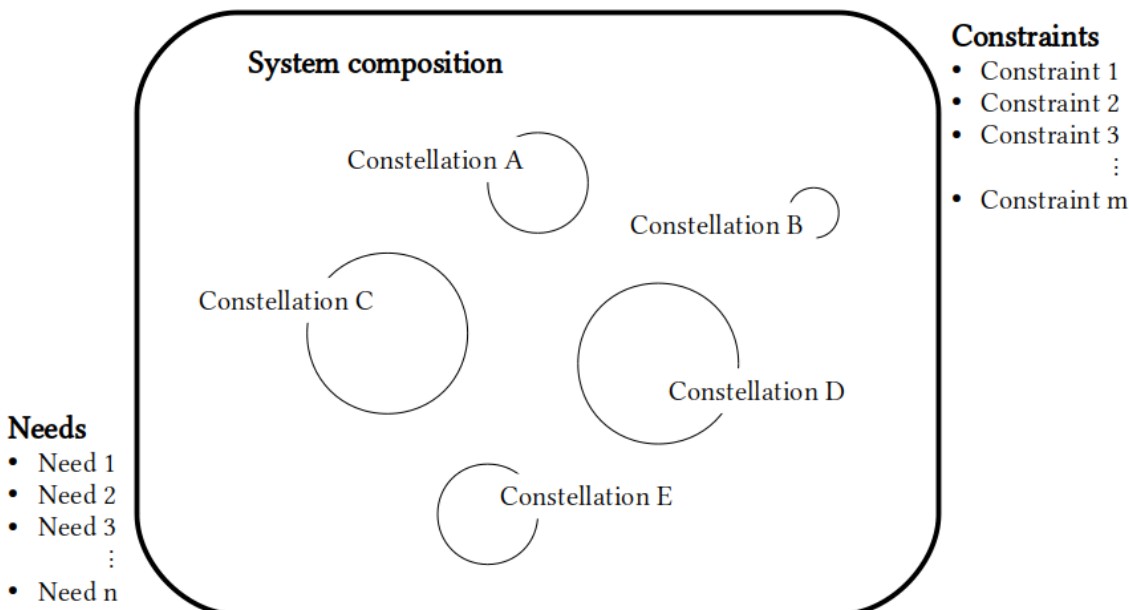

**Figure 1.** Service-provision system composed of constellations with solutions in its environment of
societal needs and constraints.

*2.2. Core Concepts—System Change*

Two assumptions are at the heart of the MPA's use of *patterns* to analyse transition pathways:
(1) That all transformative change can be captured in a limited number of patterns, and (2), that such
patterns are useful units of analysis.

To the former can be adduced that the MPA patterns are not the result of empirical induction,
rather they are theoretical constructs that—by design, and given the concepts introduced earlier—cover
all cases as a matter of definition. Of course, this in itself is not a laudable or illuminating
accomplishment. A division of all change into that which involves animals and that which does
not covers all cases also—without being very helpful. In other words, completeness is a desirable and
comforting design criterion but no guarantee for usefulness.

In defence of the latter assumption, that patterns are useful units of analysis, can be argued that
they provide a natural—in the sense of suggested by the conceptual framework—intermediate scale
between the micro level where all change is considered and the macro level of whole-of-pathway
classifications. Patterns naturally pick out this intermediate scale as they are defined in terms of
constellation change.

What then, are the patterns? Patterns are distilled straightforwardly from the constellation concept
as follows.

Patterns of Transformative Change

First, consider one constellation in isolation, but embedded in an environment of societal needs and constraints. This environment may—and will—contain other constellations, more on this shortly.

> ### Adaptation
>
> A pattern of adaptation is said to occur if the constellation changes its functioning, whether that be through adoption or phasing out of a solution or through institutional changes such as new organisations emerging. As a consequence of adaptation, a constellation's power may increase, decrease or stay the same, whatever the case may be.

Then, consider at least two constellations meeting a partially or completely overlapping set of needs.

> ### Empowerment
>
> A pattern of empowerment is said to occur if one constellation gains power at the expense of another. This may be a simple process of one set of solutions out-competing the other, for example because it operates or offers services at lower (societal) cost. It may also be a process where one constellation meets additional, higher order, needs—ecological health, equity, independence—thus having a loftier allure, if perhaps at an initially higher monetary cost.

In any one adaptation pattern, only one constellation changes—which can be aptly referred to as the protagonist. In the following, the phrasing of a pattern 'acting upon' a constellation will also be used. Adaptation only acts upon a protagonist constellation. Empowerment, by contrast, acts upon a protagonist—the constellation gaining power in the process—*and* an antagonist—who loses power. In other words, for every empowered constellation, another is *dis*empowered.

Two notes on system boundaries:

1.  Note that this definition is in a sense flexible and robust with respect to where the analyst cares to draw the system boundaries. If the analyst chooses the system boundaries so small that the entire system is composed of precisely one constellation—the smallest scale at which the MPA can meaningfully be applied—then *all* patterns are adaptations. If a closer inspection of the environment of that protagonist would reveal that a certain constellation there is gaining or losing power as the protagonist is losing or gaining it, this would indicate that the boundaries have been drawn too narrow and that one is actually observing an empowerment with the antagonist currently outside of the system boundary.
2.  Note that it is possible to 'abuse' this flexibility for analysing very large and encompassing systems. For example, analysing a decarbonisation transition would involve several service-provision systems (energy supply, housing, mobility), themselves composed of one or more constellation (per definition). It is then possible to analyse this doubly-composite system by frame-shifting the analysis, turning service-provision systems into constellations as it were. Empowerments internal to the systems-as-constellations now appear as adaptations while the solution-level of analysis is obscured.

In summary, the boundaries of a service-provision system are to a considerable extent an analytical choice, but constellations are 'real'.

The analytical flexibility of system boundary allows the distillation of another pattern, or rather pattern couple. Though the MPA refrains from conceptualising the *agents* of change, considering where the imperative for the pattern came from provides a useful distinction. What does this mean? The Adaptation and Empowerment patterns as described, implicitly suggest that they are enacted from *within* the system and for reasons related to the system or the needs it meets. In other words, these

patterns describe 'organic', internally-induced change. In many empirical cases, however, systemic change is due to an intervention—change agents from outside the system, governments, invading nations, may, for reasons not actually pertaining to the system in question, impose a new order. Let the pattern describing such dynamics be called—perhaps unimaginatively—*intervention*. Note that in de Haan and Rotmans [18], this pattern was called—imaginatively, by contrast—reconstellation. We, the present authors, began using 'intervention' to avoid the overly technical ring of 'reconstellation'. There is no conceptual difference, 'reconstellation' and 'intervention' are synonyms as far as the MPA is concerned. Analysts are of course free to choose which word they prefer. As the dynamics *resulting* from an intervention would still conform to either an adaptation or empowerment, it is apt to speak of empowerment or adaptation *by* intervention:

> **Intervention**
>
> A pattern of intervention is said to occur if an adaptation or empowerment is due to an imperative from outside the system boundaries. This may entail that the agent(s) of change operate from outside the system—e.g., a higher tier of government or other nation. The motivation of the *intervention* need not at all pertain to the system in question, as for example in the case of nations liberalising economies or wars.

### 2.3. MPA Manual

The step plan we are about to suggest should be taken as just that, a suggestion. The logic of the MPA should be robust enough to 'correct' any artefacts produced by a not-by-the-book application. Other, similar, step-by-step plans for MPA analyses can be found in (van Raak and de Haan [26], pp. 48–49 with schematic, Fig. 3.1, on p. 50), elaborately in (Van Raak [25], pp. 92–96) and in (de Haan [17], Book III, pp. 9–11). It is important to note that this manual is for applying the MPA to a past, or possibly ongoing transition. The MPA is in fact eminently suited to investigate prospective or desired transition—through the analysis of possible future pathways—but this is not what this manual is for. Figure 2 provides a schematic summary of the manual.

#### 2.3.1. Demarcation

Demarcation needs to be done in three senses:

Functional—What service-provision system is under analysis, that is, what set of related societal needs is under examination?

Temporal—What is the smallest period containing the developments that are the reason for the analysis? If at all possible, choose the initial temporal demarcation such that this period is bounded before and after by a periods of relative stability. Include these boundary periods.

Spatial—Or jurisdictional. What is the spatial 'envelope' of the dynamics? Where does the transition happen? A useful demarcation can generally be found by looking at the smallest jurisdiction under which the system falls. Consequently, the spatial demarcation will typically be a national, urban, regional, etc. *governance* area.

#### 2.3.2. Decomposition

Level 1—System into Constellations.

Identify and delineate the constellations the system is composed of. This is arguably the most important and challenging aspect of the analysis. Four kinds of consideration can help with this. Without implying hierarchy or sequentiality, one can consider the:

Subsets of needs—For example, different health care needs lead to different constellations (mental health, general practice, etc.) and water management systems often feature separate sanitation and drinking water supply constellations.

Similar solutions—Institutional coherence is both reinforced by and favours similar solutions. Thus centrally generated electricity based on various sources (coal, gas, nuclear) usually live in the same constellation, and, while each medical specialty is obviously special, specialists operate similarly in the sense of professional accreditation, the administrative trajectory their patients undergo and in their reliance on the hospital as their workspace.

Ownership and service model—One manifestation of institutional coherence of constellations is the similarity in service and ownership models within a constellation. For example, in mobility systems, constellations can often be identified by distinguishing between collective versus individual services (i.e., public transport versus individually owned and operated vehicles), and between public versus private service models (further separating normal public transport from privately operated modes such as taxis).

Development—Sometimes, looking at the temporal development of a constellation sheds light on its delineation. Some constellations are transient, becoming obsolete or getting absorbed into others. The societal needs met by one constellation in a particular context, may— because of historical contingencies—be met by two or several in another.

### Level 2—Constellations into Solutions

Per constellation, identify the solutions it implements to meet societal needs. This identification likely already partly emerged from the decomposition into constellations. Sometimes, it is easy to recognise the solutions in a constellation (e.g., trains, buses, trams in public transport), sometimes less so. Technologies and infrastructures may appear good candidates for solutions, but it is not always one-technology-one-solution. (Is the solution the power plant or is it 'grid-based, coal-fired, electricity generation and distribution'? The latter.) Some solutions involve technology but incidentally so (dental health practice, classroom teaching), some per definition hardly at all (walking as a mode of transport, 'talking therapy' in mental health as earlier mentioned).

### 2.3.3. Dynamics

Constellation timelines—For each constellation, prepare a timeline spanning the entire period identified in the temporal demarcation phase. Gather an overview of key events and the changes it brought with dates and sources.

Pattern chains—Divide each constellation-timeline into a chain of patterns. First, delineate the patterns, i.e., what sets of change belong together, where does a pattern begin and end. Then, identify if it concerns an adaptation, an empowerment or an intervention. If an empowerment is identified, find the corresponding antagonist or protagonist constellation in another timeline, i.e., what is *dis*empowered as a consequence of this empowerment or the other way around.

Conditions—Identify the conditions under which each pattern eventuated. For example, what societal needs were not met? What were the constraints?

Connecting timelines—Identify the interactions that connect the timelines of different constellations. Typically, this would mean tracing the empowerment patterns identified when unravelling pattern chains. Another important way constellation-timelines may become connected is when 'loops are closed'. For example, where the services or waste of a solution in one constellation become a prerequisite for the functioning of a solution in another, such as in the case of using recycled wastewater (sanitation constellation) for non-potable water supply (water supply constellation).

Recognising pathways—The overall case may now 'fall apart' in separate sets of pathways, into connected constellation timelines. This need not be the case of course, but, if it is, separating the pathways may allow a clearer presentation of the results or provide a sub-case for more detailed analysis (see below).

Pivot periods—Certain periods may feature a cascade or a 'coming together' of several patterns across several constellation timelines. Such a period is a pivot period of a transition and an obvious candidate for a more detailed analysis (see below).

Note that any of these phases may yield insights that necessitate a revisiting of an earlier phase. For example, unravelling the pattern chains in Dynamics phase may lead to a revision of the Decomposition of the system into constellations, or of the functional Demarcation altogether. In addition, the dynamics can be analysed at two levels of detail. At the level of patterns changing constellations, or at a more detailed level where the changes are more precisely traced in terms of individual solutions being adopted or phased out.

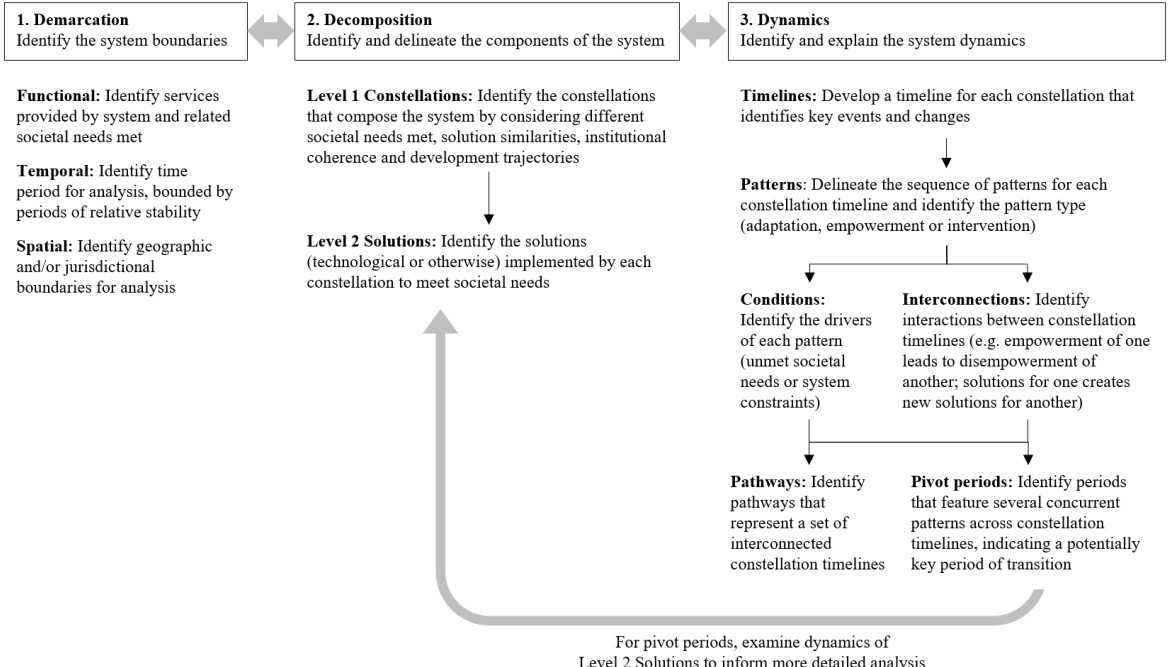

**Figure 2.** Method for applying the MPA.

Patterns can be described compactly in tabular form (see Sections 3.3.3 and 3.4.3 for examples). At a minimum, a pattern description should entail the following:

- *Pattern type* (empowerment, adaptation—possibly by intervention), the *constellation(s) affected* by the pattern and *when* it was active.
- The conditions under which the pattern occurred, that is, the relevant needs and constraints present during the pattern's workings.
- For a detailed analysis, a pattern description would also include the affected *solutions* of each affected constellation.
- The consequences of the pattern for the constellation(s), that is, *which ones increased or decreased their power*, possibly noting whether one gained power at the expense of another. In the pattern descriptions in Sections 3.3.3 and 3.4.3 a notation of [ + ] and [ − ] is used to signify gaining or losing power. This is used in similar fashion for the uptake and phasing out of solutions, which is the detailed analysis of constellations gaining and losing power. If a constellation or solution emerges afresh, this can be denoted by [ 0+ ], while a constellation or solution being affected without gaining or losing power can be denoted by [ ∼ ].

## 3. Demonstration of Empirical Application

The story of Melbourne's water system has evolved over the last two centuries since the first arrival of Europeans in response to emerging social, political, climatic, ecological and economic drivers. This evolution has been characterised by steady and then accelerated population growth, economic fluctuations, periodic drought and growing community aspirations for a clean, healthy urban environments. Significant infrastructural and institutional changes have resulted for the water

system—over time, new system services have been added and the approach to deliver some of those services have been adapted. These changes are particularly marked since 1995 when pollution control was given greater priority and the city experienced extended drought. A commitment to integrated water cycle management and liveability outcomes emerged over this period, enabled by the integration of decentralised and nature-based technologies with traditional centralised infrastructure [27,40].

This section uses the MPA to analyse Melbourne's water system changes, following the manual set out in Section 2.3, to demonstrate application of the method and highlight its value as an analytical approach for explaining the broad patterns of system change (Section 3.3) as well as the detailed uptake and phasing out of system solutions (Section 3.4).

### 3.1. Data Collection and Analysis

The analysis of Melbourne's water system developments took a qualitative single-case study approach [41]. A summary of the empirical methods adopted is presented here, while a more detailed account of the primary data collection techniques and protocols, as well as secondary data sources, can be found in Ferguson et al. [40]. Secondary data was sourced from historic records, published literature, policy materials, organisation reports and media reports. Primary data was collected through interviews with 29 informants with an extended career in the water sector, including executives, managers and project officers from water utilities, state government agencies, local municipalities, academia and private consultants. Two interview techniques were adopted [42]: oral histories elicited detailed personal accounts of Melbourne's water system changes through in-depth free flowing discussion and semi-structured interviews explored key drivers, barriers and enablers of the significant changes experienced in recent decades. The first phase of analysis created a chronological narrative of changes in Melbourne's water system since European settlement, which involved triangulating data sources to develop converging narratives. The second phase of analysis applied the MPA to narrative to explain key periods of Melbourne's water evolution, the patterns that have driven these shifts and the dynamics of the uptake and phasing out of specific solutions.

### 3.2. Narrative of Melbourne's Water System Changes

Prior to the arrival of European settlers, Aboriginal people relied on the Yarra River as an important source of freshwater; a feature that was the key attraction for establishing the colony of Victoria in 1835 where Melbourne is now located. Early settlers also fetched water from the Yarra River with buckets and, a few years later, through private operators pumping and carting water to sell door-to-door. Waste was dumped in the streets, but the population was low so impacts were limited [43,44].

The Gold Rush of the 1850s led to a rapid growth in population, industry and wealth for Melbourne [45]. The Yarra River became highly polluted and was no longer a reliable water source, leading to strong community demand for better quality and quantity of water supplies [44]. The accumulating wealth meant money became available to provide a centralised resource through the construction of Yan Yean Reservoir to the north of Melbourne in 1857. The connected supply network expanded across Melbourne, gradually reducing the role of private pumping and carting operations.

By the 1880s, public health had become a significant problem with regular outbreaks of waterborne diseases such as typhoid and cholera [45]. Solid waste and wastewater were dumped in street channels and an unreliable nightsoil system was in place. Melbourne's poor international reputation was characterised by its pseudonym 'Smellbourne' and public demand for a solution was strong. In response, a centralised sewerage network was progressively built from the 1890s and the Western Treatment Plant in Werribee was operational from 1897 [46].

Both the water supply and sewage networks expanded throughout the 1900s as the city grew. A second large wastewater treatment plant in Melbourne's east was constructed in 1975 and onsite septic tanks have been gradually replaced with sewage networks (a backlog program is still ongoing). Periodic water shortages were addressed by constructing additional water supply reservoirs, the

largest being the Thomson Reservoir, which, upon completion in 1984, was heralded as having 'drought-proofed' Melbourne.

The baby boom and immigration that followed World War II led to a rapid expansion of Melbourne's footprint, as people maintained their desire for large suburban blocks [45,47]. Vast areas of land were covered with impervious materials and stormwater inundation became a big issue [46]. A network of pipes and drainage channels were constructed (separately from sewers) to convey stormwater runoff to local waterways, before being discharged to Port Phillip Bay. Runoff from major storm events was directed to flow overland along roads, easements and designated floodways towards the receiving creeks and rivers.

Modern environmental values emerged during the late 1960s and 1970s, and people's care for ecological health and amenity of urban water bodies highlighted vulnerabilities associated with the water system features developed to date. Point sources of pollution, such as industrial effluent, were largely addressed with the widespread availability of sewerage networks and regulation of waste discharges. However, the diffuse nature of stormwater was more difficult to manage and continued as a key source of pollution for receiving waterbodies.

Research into alternative stormwater quality management infrastructure began in the mid 1990s, developing green technologies (e.g., vegetated swales, biofilters, wetlands) that could replace concrete-lined drains. Over the following two decades, policies, regulations and technologies continued evolving to reflect the strengthening priorities that both stormwater quality and quantity need to be addressed. Floodplain zoning also emerged as an alternative approach to managing water flows in storm events that did not rely purely on rapidly conveying large volumes of water away from vulnerable areas.

Opportunities for improved environmental outcomes were also pursued through alternative wastewater management approaches, such as recycling to reduce the impacts of polluted discharges to receiving environments. This was considered particularly promising after the onset of the Millennium drought in 1997, when water reservoir levels were dwindling. While a widespread water saving campaign successfully changed consumer behaviour and increased the efficiency of water use by business and industry, additional water resources were also required as the drought intensified. The desire for increased resilience against climate change led to the adoption of a 'portfolios' approach that could tap into a diversity of supply sources, including recycled wastewater and harvested stormwater. The focus of these innovations has been on supplying 'fit-for-purpose' water that is lower than drinking quality standard but suitable for toilet flushing and irrigation of private gardens, parklands and food crops. These initiatives were driven by households and private companies, in addition to the public utilities. Three major infrastructure projects were also commissioned in 2007 to augment water supplies, including Australia's largest desalination plant and the North-South pipeline, designed to convey water to Melbourne from the Goulburn River system in rural Victoria [48].

The Millennium Drought ended in 2010 and was followed by two years of record levels of rainfall and flash flooding became a significant community concern. These patterns, along with a growing awareness of the impacts of climate change and ongoing urbanisation has led Melbourne's water sector to focus on the role of water in enhancing urban liveability by providing amenity, recreational and ecological benefits [49,50].

### 3.3. MPA Case Analysis: Level 1—Broad

3.3.1. Demarcation

The water system that services the greater metropolitan region of Melbourne (spatial demarcation), as described in the narrative above, delivers a number of important functions to society. It manages all aspects of the water cycle to supply water, provide sanitation, drain stormwater and protect people and property from flooding, mitigate pollution and hydraulic impacts to protect the ecological health of waterways, and create cool and amenable blue and green environments for passive and active recreation (functional demarcation).

The narrative of how these different services evolved over the case study is delineated into time periods of relative stability that are characterised by particular drivers and developments (temporal demarcation). These periods are tabulated in Table 1.

3.3.2. Decomposition

Seven distinct constellations can be identified in the case narrative for Melbourne's water system, as defined in Table 2. While each constellation changes over time in response to the pressures in the overall system, they remain institutionally coherent in that they have a set of institutional structures that are qualitatively different to other constellations, although there is some overlap in the types of solutions that make up different constellations.

**Table 1.** Key periods and conditions of Melbourne's water management history.

| Period & Definining Character | Needs | Constraints |
|---|---|---|
| [1835–1850] Early European Settlement<br>[1851–1880] During the Gold Rush<br>[1881–1935] Dealing with Smellbourne<br>[1936–1965] Post-War Urban Expansion<br>[1966–1995] Emerging Environmentalism<br>[1996–2015] Liveability and Resilience | [1835–1890] Potable and non-potable water supply due to population and economic growth<br>[1845–1930] Protection of public health due to water-borne diseases<br>[1925–1965] Protection of property due to stormwater runoff<br>[1990–2015] Urban space for amenity, recreation and environmental health | [1997–2010] Lack of water available due to drought<br>[2005–2015] Lack of resources such as nutrients |

**Table 2.** System constellations in Melbourne's evolving water system.

| Constellation | Abbr. | Service Aim (in Relation to Needs Met) | Institutional Characteristics | Solutions [1] |
|---|---|---|---|---|
| «Private water management» | «PWM» | Provide reliable, secure, equitable and efficient water system services while encouraging autonomy and market competition | Service provided by households, communities and private companies, supported by government in a regulative and/or enabling role | • Septic tanks<br>• Water efficient appliances<br>• Onsite treatment systems<br>• Rainwater tanks<br>• Sewer mining<br>• Wastewater recycling systems |
| «Public water supply» | «PWS» | Provide a reliable, secure, equitable and efficient supply of water for consumptive purposes | Service provided by government, with central management and tight regulations | • Reservoirs and dams<br>• Treatment plants<br>• Transfer pipelines<br>• Reticulation networks<br>• Demand management |
| «Public sanitation servicing» | «PSS» | Protect public health by efficiently and safely removing human and trade waste while minimising environmental impact | Service provided by government, with central management and tight regulations | • Nightsoil collection<br>• Sewage collection pipe networks<br>• Wastewater treatment plants<br>• Energy recovery<br>• Nutrient recovery * |

**Table 2.** *Cont.*

| Constellation | Abbr. | Service Aim (in Relation to Needs Met) | Institutional Characteristics | Solutions [1] |
|---|---|---|---|---|
| «Fluvial & coastal flood management» | «FCM» | Protect people and property from coastal and fluvial flooding | Service provided by government, with central management, tight regulations and a management role for private property owners | • Dams<br>• Parks and other open space<br>• River and stream modifications<br>• Emergency response plans<br>• Floodplain zoning<br>• Real-time control systems *<br>• Dykes and pumps * |
| «Public stormwater conveyance» | «PSC» | Drain stormwater runoff away from people and property safely and efficiently | Service provided by government, with central management and tight regulations | • Concrete-lined channels<br>• Drainage pipe networks<br>• Overland flow paths<br>• Retarding basins |
| «Green water management» | «GWM» | Provide water management services integrated with functioning ecosystems and aesthetic landscapes in natural and built environments | Services provided by diverse range of actors, including government, community and the private sector | • Wetlands, ponds and biofiltration treatment systems<br>• Water detention systems<br>• Rainwater and stormwater harvesting systems |
| «Waterbody access» | «WBA» | Make waterbodies environmentally healthy and accessible for non-consumptive purposes | Services provided by diverse range of actors, including government, community and the private sector | • Clean coastal, estuarine and inland waters<br>• Pollution controls<br>• Hydraulic controls<br>• Walking and cycling trails<br>• Ferry routes and shipping lines *<br>• Fishing boat ramps |

[1] The solutions that form constellations vary over time as they evolve in response to system pressures. This list is indicative of the solutions that were part of Melbourne's water system at different times, whether or not they are explicitly identified in the narrative. The exception is solutions marked with an *, which could belong in a similar constellation for another system and are presented to demonstrate a wider range of solutions that could contribute to such a constellation's functioning.

3.3.3. Dynamics

Figure 3 represents the changes in each constellation identified in Table 2, the patterns that drove these changes and the conditions under which the patterns manifested. The connections between constellations are indicated with an arrow (i.e., under the empowerment pattern).

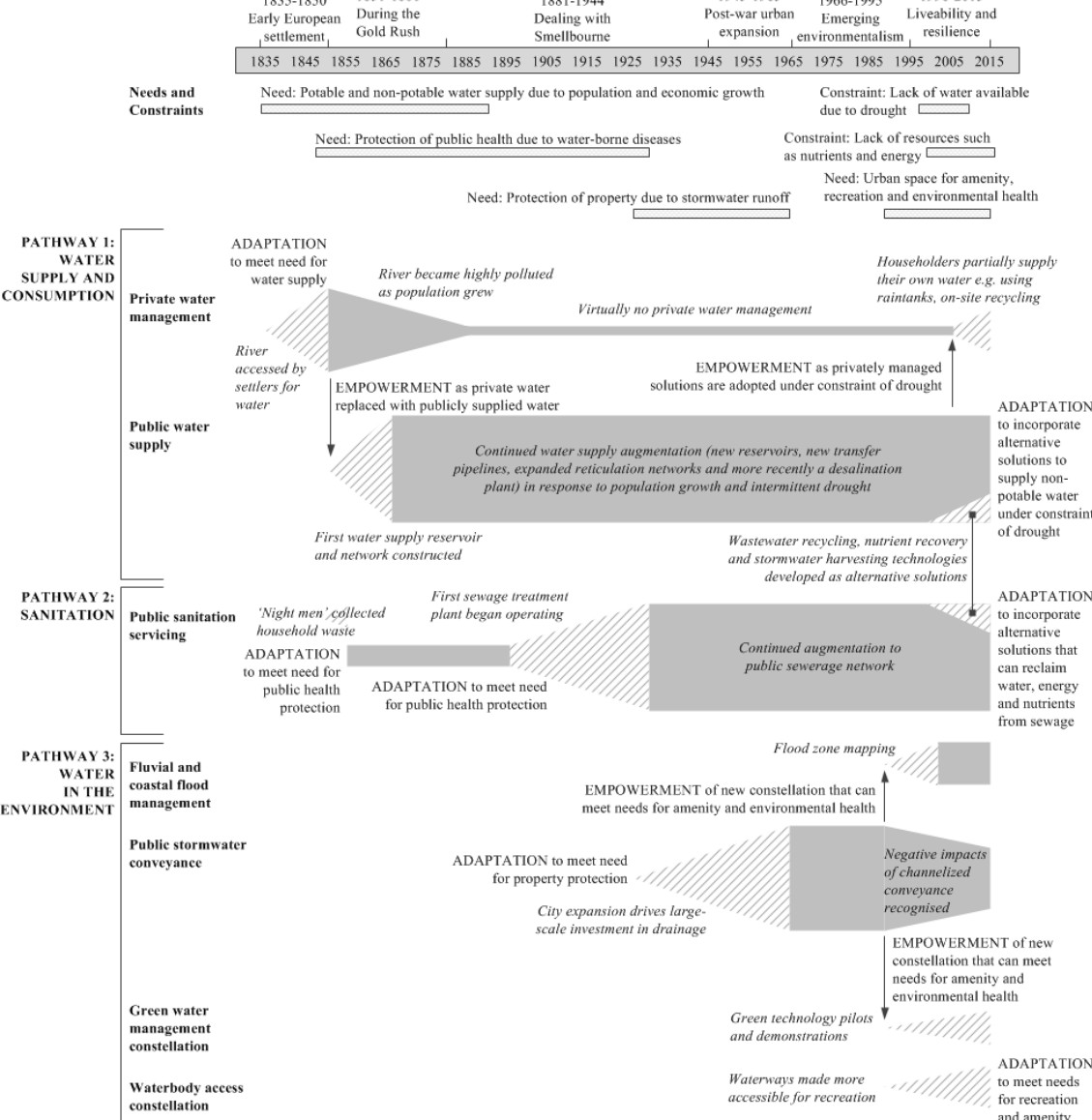

**Figure 3.** MPA analysis of Melbourne's water system. The bands represent the temporal development of the constellations, with the size in the vertical direction indicative of a constellation's relative power. Dashed areas indicate a pattern acting, i.e., change. Arrows point at connections between constellations under influence of a pattern.

Analysing across all the constellations of Figure 3, the dynamics of Melbourne's water system over the case study period can be understood as three individual but intersecting pathways of change. Water supply and consumption has shifted from small-scale private provision to a large-scale public system underpinned by centralised infrastructure. More recently, the supply system has incorporated decentralised and lower quality water sources such as recycled wastewater and harvested stormwater, which also reopens opportunities for private sector and household involvement in supplying water. The provision of sanitation services has expanded over the case study period, with a large-scale

public system gradually replacing privately operated nightsoil services and household septic tanks. Management of water in the environment has changed dramatically over time, as early concerns for conveying stormwater away from people and property needed balancing with concerns for environmental health, recreational opportunities and urban amenity.

These pathways and their constituent patterns are now described with the MPA.

### Analysis of Pathway 1—Water Supply and Consumption

This pathway involves the «Public Water Supply» and «Private Water Management» constellations. Each emerged in response to the basic needs for potable and non-potable water, which were growing as a consequence of European settlement and development of economic activity. As Melbourne rapidly urbanised, «Public Water Supply» became dominant, while «Private Water Management» could no longer meet the growing population's need for water supply. «Public Water Supply» continued as a dominant constellation, with augmentations including new reservoirs and more recently desalination. «Private Water Management» was virtually non-existent until the Millennium Drought constrained the availability of public water supply and drove individual households to partially supply their own water through rainwater tanks, etc.

| Adaptation | of «Private Water Management» | [ 1835—1850 ] |
|---|---|---|

**Conditions**

| *Needs* | | *Constraints* |
|---|---|---|
| [ - ] | Potable water | ø |
| [ - ] | Non-potable water | |

**Constellations**

| « PWM » | [ + ] |
|---|---|

**Synopsis**

Pioneer settlers establish basic water supply facilities ranging from directly collecting water from the Yarra River with buckets and later by private operators pumping and carting water sold door-to-door.

| Adaptation | of «Public Water Supply» | [ 1850—1880 ] |
|---|---|---|

**Conditions**

| *Needs* | | *Constraints* |
|---|---|---|
| [ - ] | Potable water | ø |
| [ - ] | Non-potable water | |

**Constellations**

| « PWS » | [ + ] |
|---|---|

**Synopsis**

Establishment of a reticulation system supplying water from the Yan Yean reservoir to Melbourne.

| Empowerment | of «Public Water Supply» at the expense of «Private Water Management» | [ 1835—1850 ] |
|---|---|---|

**Conditions**

| *Needs* | | *Constraints* |
|---|---|---|
| [ - ] | Potable water | ø |
| [ - ] | Non-potable water | |

**Constellations**

| **« PWS »** | [ + ] | **« PWM »** | [ - ] |
|---|---|---|---|

**Synopsis**

The services from the **« PWS »** constellation soon make those of the **« PWM »** obsolete.

| Adaptation | of «Public Water Supply» | [ 2002—2010 ] |
|---|---|---|

**Conditions**

| *Needs* | | *Constraints* |
|---|---|---|
| [ - ] | Potable water | [ 0+ ] Water scarcity |
| [ - ] | Non-potable water | |

**Constellations**

**« PWS »**

**Synopsis**

Public Water Supply « PWS » responds to constraint of water scarcity, particularly because of the Millennium Drought and a desalination is adopted to the meet unmet need for potable and non-potable water supply.

| Adaptation | of «Public Water Supply» | [ 2004—ongoing ] |
|---|---|---|

**Conditions**

| *Needs* | | *Constraints* |
|---|---|---|
| [ - ] | Non-potable water | [ 0+ ] Water scarcity |

**Constellations**

| **« PWS »** | [ ~ ] |
|---|---|

**Solutions**

| [ ~ ] | Reticulated supply of reservoir water |
|---|---|
| [ 0+ ] | Treatment and reticulated supply of recycled sewage |

**Synopsis**

« Public Sanitation Servicing » enables treatment of recycled of waste water up to human-contact standards (see Pathway 2), enabling « PWS » to implement it as a new water source.

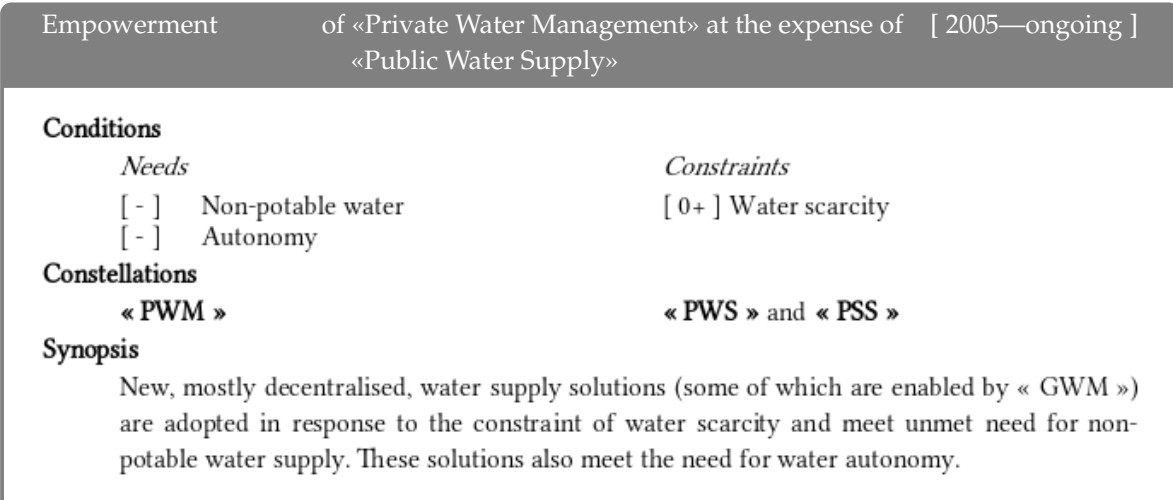

Empowerment of «Private Water Management» at the expense of «Public Water Supply» [ 2005—ongoing ]

**Conditions**

| *Needs* | *Constraints* |
|---------|---------------|
| [ - ]  Non-potable water | [ 0+ ] Water scarcity |
| [ - ]  Autonomy | |

**Constellations**

« PWM »  « PWS » and « PSS »

**Synopsis**

New, mostly decentralised, water supply solutions (some of which are enabled by « GWM ») are adopted in response to the constraint of water scarcity and meet unmet need for non-potable water supply. These solutions also meet the need for water autonomy.

Analysis of Pathway 2—Sanitation

This pathway involves «Public Sanitation» as the only constellation. Sanitation as a public service emerged soon after settlement, initially limited to small-scale collection and disposal of waste. As Melbourne's population rapidly increased, disease (cholera) and odour drove the installation of centralised sewer system to meet the need for public health protection. «Public Sanitation» continued as a dominant constellation, with augmentations to the treatment system and expansion of the sewer network. In the 2000s, the need to protect the environmental health of downstream waterways drove new solutions to treat sewage to a higher standard, and the potential to use this as an alternative water resource became apparent under the water supply constraints of the Millenium Drought. More recently, the potential to recover energy and nutrients from treated sewage has further driven the adoption of new solutions.

Adaptation of «Public Sanitation Servicing» [ 1850—1855 ]

**Conditions**

| *Needs* | *Constraints* |
|---------|---------------|
| [ - ]  Public health | ø |

**Constellations**

« PSS »

**Synopsis**

Brief adoption of a system where 'night men' collected solid waste.

Adaptation of «Public Sanitation Servicing» [ 1890—1930 ]

**Conditions**

| *Needs* | *Constraints* |
|---------|---------------|
| [ - ]  Public health | ø |

**Constellations**

« PSS »

**Synopsis**

Establishment of a centralised sewer system from the 1890s onwards meeting the public health need.

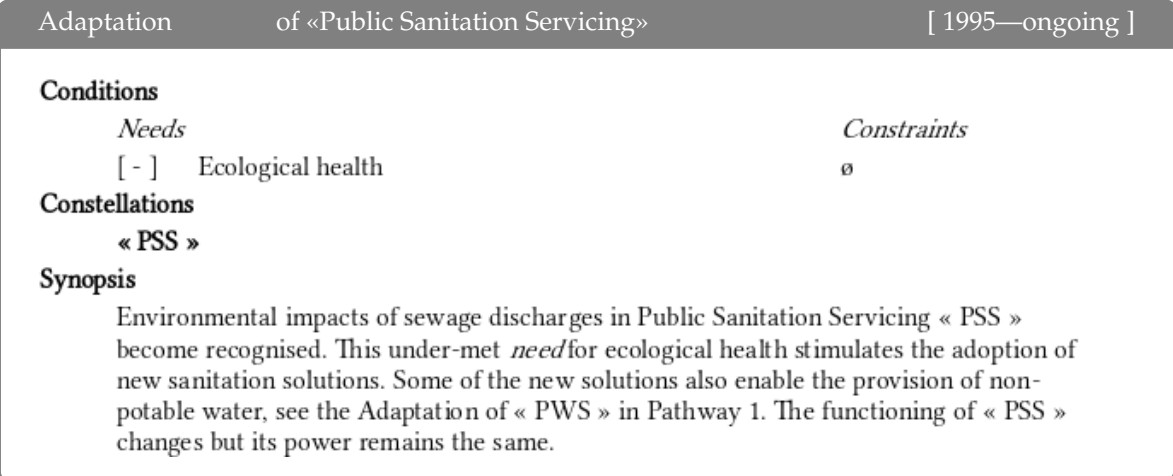

Analysis of Pathway 3—Water in the Environment

This pathway involves the «Public Stormwater Conveyance», «Water Body Access», «Green Water Management» and «Fluvial & Coastal Flood Management» constellations, reflecting the diverse ways in which water and the urban landscape interconnect. «Public Stormwater Conveyance» initially emerged to primarily meet the need for property protection (and, to a lesser degree, the need for public health and amenity by avoiding stagnant water and mud) through centralised drainage infrastructure. More recently, the needs for recreation and amenity have driven the emergence of «Water Body Access», for example installing bicycle and walking trails to make waterways publicly accessible. As the need for environmental health protection became more pressing and unmet by «Public Stormwater Conveyance», «Green Water Management» emerged to drive the installation of nature-based technologies that remove pollutants from water, as well as enhance urban amenity. «Fluvial & Coastal Flood Management» also began incorporating managerial and regulative measures to deal with floods, rather than only «Public Stormwater Conveyance».

| Adaptation | of «Public Stormwater Conveyance» | [ 1926—1965 ] |
| --- | --- | --- |

**Conditions**

    *Needs*                                  *Constraints*

    [ - ]    Property protection             ø

**Constellations**

    « PSC »

**Synopsis**

    Emergence of « PSC » constellation, implementing storm sewers and similar centralised infrastructure to meet the need for property protection – soon meeting the needs of an expanding Melbourne. To a lesser degree « PSC » also meets public health (stagnant water) and amenity (avoiding mud and puddles) needs.

| Adaptation | of «Water Body Access» | [ 1926—1965 ] |
|---|---|---|

**Conditions**

| *Needs* | *Constraints* |
|---|---|
| [ - ]　Recreation | ø |
| [ - ]　Amenity | |

**Constellations**

« WBA »

**Synopsis**

Emergence of « WBA » constellation, first starting to meet recreation and amenity needs. Subsequent partial meeting of environmental health need.

| Empowerment | of «Green Water Management» at the expense of «Public Stormwater Conveyance» | [ 1990—ongoing ] |
|---|---|---|

**Conditions**

| *Needs* | *Constraints* |
|---|---|
| [ - ]　Ecological health | ø |

**Constellations**

| « GWM »　[ + ] | « PSC »　[ - ] |
|---|---|

**Synopsis**

Uptake of « GWM » solutions to deal with adverse side effects of functioning of « PSC » - in particular the diffuse source pollution caused by stormwater run off entering waterways.

| Empowerment | of «Fluvial & Coastal Flood Management» at the expense of «Public Stormwater Conveyance» | [ 1990—ongoing ] |
|---|---|---|

**Conditions**

| *Needs* | *Constraints* |
|---|---|
| [ - ]　Ecological health | ø |

**Constellations**

| « FCM »　[ + ] | « PSC »　[ - ] |
|---|---|

**Synopsis**

Increased environmental awareness suggests different ways to manage nuisance flooding, leading to the emergence and slight empowerment of the new « FCM », using managerial and regulative measures to deal with floods.

*3.4. MPA Case Analysis: Level 2—Detailed*

In this part, the analysis will be deepened by focussing on a more specific—shorter—period of the case-study time line presented in Part 1. Here, the analysis will go to what was called Level 2 in the manual. In other words, in addition to analysing in terms of changing constellations within the larger system, this part will also include changing solutions within those constellations.

3.4.1. Demarcation

The spatial demarcation is the same as for the Case Analysis Part 1, that is, metropolitan Melbourne. The functional demarcation is also the same, though the focus is restricted to those constellations implicated in the dynamics related to the Millennium Drought. The temporal demarcation is now limited to the most recent two decades from 1995 to 2015.

### 3.4.2. Decomposition

The Level 1 decomposition can be imported from Part 1 directly and as the functional demarcation remains the same, all identified constellations are considered—with the exception of «Waterbody Access» and « Iood Management». For Level 2 decomposition, the constellations considered need to be 'taken apart' into the solutions they implement. The constellations and the solutions they have implemented are listed below. Note that all solutions listed for a given constellations were implemented during the entire period considered—some were adopted, some got phased out—these are precisely the dynamics to unravel.

**«Public Water Supply»**

- Reticulated supply of reservoir water
- Demand management
- Reticulated supply of desalinated water

**«Public Sanitation Servicing»**

- Sewage treatment plants (to environmental standards)
- Treated sewage outfalls
- Recycled sewage plants (to human contact standards)
- Sewer mining schemes

**«Private Water Management»**

- Small-scale sanitation technologies (e.g., septic tanks, composting toilets)
- Small-scale supply technologies (e.g., rainwater tanks, on-site greywater reuse, treated stormwater reuse)
- Small-scale technologies to improve efficiencies (e.g., showerheads)
- Industrial-scale reuse systems (e.g., sewer mining, on-site recycling)

**«Green Water Management»**

- Pollution removal (e.g., wetlands, ponds, biofilters)
- Stormwater detention
- Stormwater capture and storage

**«Public Stormwater Conveyance»**

- Overland flow paths
- Channelised waterways
- Concrete kerb and channels
- Pipe network

### 3.4.3. Dynamics

*Constellation Timeline*

The changes in the period of 1995 to 2015 occur under the influence of two main conditions, (1) the emerging need for ecological health, and (2) the constraint of water scarcity imposed by the Millennium Drought. Under these unfolding conditions, all constellations under consideration implemented new solutions, including an instance of a non-technological one. As part of these dynamics, some needs became expressed more strongly and new ones emerged (e.g., autonomy, amenity). See Figure 4 for a an overview.

*Pattern Chains including Conditions*

How these conditions shaped the dynamics will now be described by breaking down the timeline into several patterns. In the description of each pattern, we will include the uptake or phasing out of solutions, under which needs and constraints this occurred. The emergence of new needs will also be indicated.

| Empowerment | of «Green Water Management» at the expense of «Public Stormwater Conveyance» | [ 1995—ongoing ] |
|---|---|---|

**Conditions**

*Needs*                                                                 *Constraints*

[ - ]     Ecological health                               ø
[ - ]     Amenity
[ - ]     Recreation
[ - ]     Autonomy
[ - ]     Non-potable water

**Constellations**

« GWM »        [ + ]                          « PSC »[ - ]

**Solutions**

[ 0+ ]     Pollution removal (e.g. wetlands,     [ ~ ]     Overland flow paths
            ponds, biofilters)
[ 0+ ]     Stormwater detention                  [ ~ ]     Channelised waterways
[ 0+ ]     Stormwater capture and storage        [ ~ ]     Concrete kerb and channels
                                                  [ ~ ]     Pipe network

**Synopsis**

Environmental impacts of Public Stormwater Conveyance « PSC » become recognised. This under-met *need* for ecological health, together with several others, stimulate the adoption of new *solutions* in « GWM ». Functioning of « PSC » stays roughly the same, and « GWM » emerging shifting the power balance slightly to the latter.

| Adaptation | of «Public Sanitation Servicing» | [ 1995—ongoing ] |
|---|---|---|

**Conditions**

*Needs*                                                                 *Constraints*

[ - ]     Ecological health                               ø
[ - ]     Non-potable water

**Constellations**

« PSS »  [ ~ ]

**Solutions**

[ ~ ]     Sewage collection pipe network

[ ~ ]     Main sewers and pumps
[ ~ ]     Sewage treatment plants (to environmental standards)
[ ~ ]     Treated sewage outfalls
[ 0+ ]     Sewer mining schemes

**Synopsis**

Environmental impacts of sewage discharges in Public Sanitation Servicing <PSS> become recognised. This under-met *need* for ecological health stimulates the adoption of new sanitation solutions. Some of the new solutions enable solutions for non-potable water supply, see the Adaptation of « PWS » below. The functioning of « PSS » changes but its power remains the same.

| Adaptation | of «Public Water Supply» | [ 2002—2010 ] |
|---|---|---|

**Conditions**

| *Needs* | *Constraints* |
|---|---|
| [ - ]     Potable water | [ 0+ ] Water scarcity |
| [ - ]     Non-potable water | |

**Constellations**

    « PWS »     [ ~ ]

**Solutions**

    [ ~ ]     Reticulated supply of reservoir water
    [ + ]     Demand management
    [ 0+ ]     Reticulated supply of desalinated water

**Synopsis**

    Public Water Supply « PWS » responds to constraint of water scarcity and meet unmet need for potable and non-potable water supply.

| Adaptation | of «Public Water Supply» | [ 2004—ongoing ] |
|---|---|---|

**Conditions**

| *Needs* | *Constraints* |
|---|---|
| [ - ]     Non-potable water | [ 0+ ] Water scarcity |

**Constellations**

    « PWS »     [ ~ ]

**Solutions**

    [ ~ ]     Reticulated supply of reservoir water
    [ 0+ ]     Treatment and reticulated supply of
recycled sewage

**Synopsis**

    New solutions in « PSS » as a result of its earlier Adaptation pattern, enable new solutions in « PWS » to respond to constraint of water scarcity and meet unmet need for non-potable water supply.

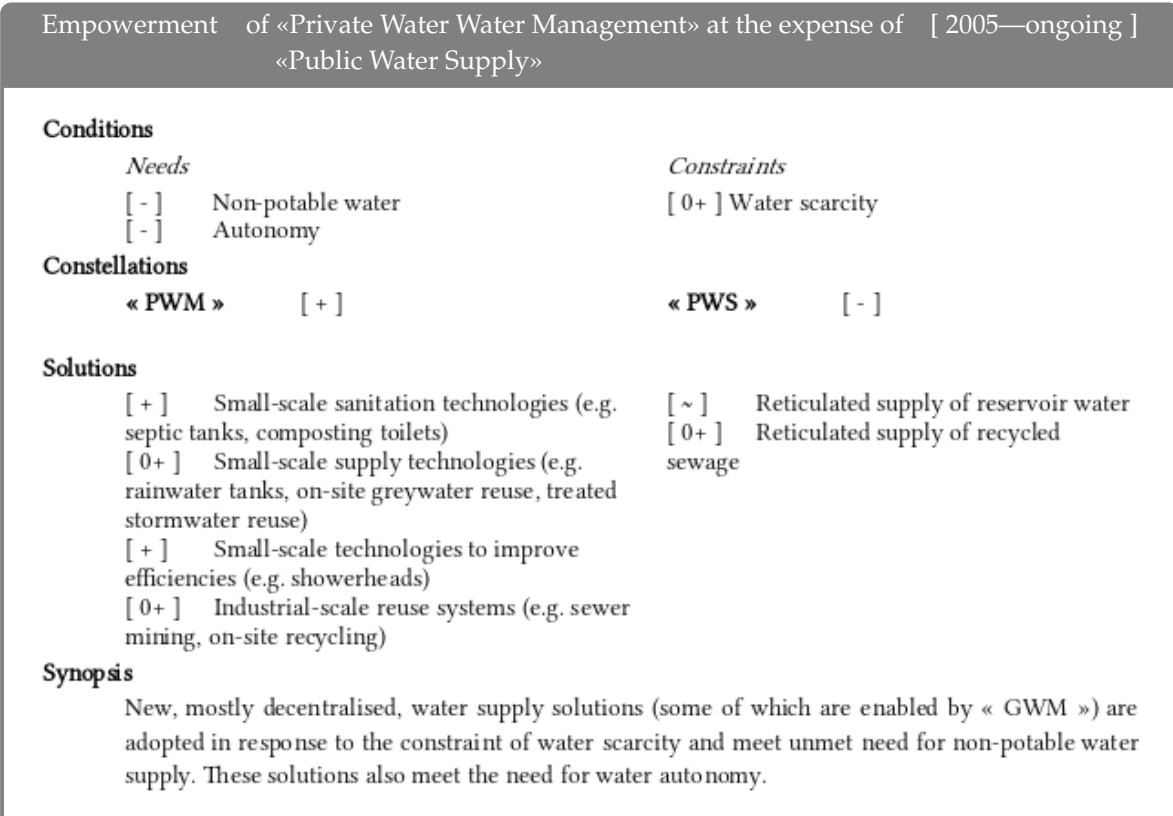

## 3.5. Case Insights Derived through MPA Application

The broad application of the MPA (Section 3.3) revealed that, up until the late 20th century, all of the constellations focused on meeting basic needs, culminating in well-functioning centralised servicing systems. From the 1990s onwards, a diversity of needs emerged, including ecological health protection and increased amenity, which led to new constellations that increased the complexity of the overall water systems servicing.

The detailed application of the MPA focused on the uptake and phasing out of individual solutions between 1995 and 2015. It showed an emerging need for ecosystem health, in combination with the driver of drought (need for water scarcity), drove a raft of patterns as identified in Section 3.4.3. In the process, some needs emerged (e.g., autonomy, amenity) or were expressed more strongly. In response, the Australian water industry has adopted the concept of liveability to express how it perceives its new role in delivering broader societal benefits as part of water system servicing. This is made evident in state, metropolitan and local policies and strategies (e.g., Victorian Water Plan, Melbourne Water doc, City of Port Phillip strategy).

The detailed MPA application revealed a further aspect of the complexification of the water system. Certain new solutions in one constellation provided inputs for other solutions in other constellations. This was seen as 'sewage recycling plants' in PSS created a resource that enabled a new solution 'reticulated supply of recycled sewage' in PWS.

Examining the current system status (from both the broad and detailed MPA application results), Melbourne's water system appears to be on a pathway towards more sustainable management approaches (e.g., emergence of green water management and various recycling and reuse solutions). The dominant servicing constellations (PWS, PSS and PSC) are in a state of flux, which arguably makes them receptive to further change such as the uptake of new and alternative solutions. A policy recommendation from this analytical result could be to reinforce these patterns to continue the empowerment of GWM and PWM, as well as their own adaptations.

## 4. Conclusions

In this article, we presented the MPA as a methodology for systematic analysis of transition pathways, with guidance on its application and demonstration on an illustrative case study of Melbourne's water system. We now conclude with some reflections on the value of the MPA as an analytical tool.

An MPA analysis is always contextualised, given the methodology's requirement to demarcate the system under analysis spatially, functionally and temporally. This gives the approach the flexibility to analyse sector-based cases and geographic delineated cases in the same terms. For example, while the illustrative case in this paper is focused on Melbourne's water system, the methodology is fully generic and could be applied to other sectoral or place-bound systems.

Depending on the case requirements, the MPA allows for both a high-level mapping of a system's evolution over time and a detailed analysis of the uptake and phasing out of specific servicing technologies and practices. In MPA terminology, the high level mapping identified what the patterns were, while the detailed analysis identified what drove those patterns more specifically, the interrelationship between patterns and the specific solutions that emerged as a result.

The modularity of MPA pattern analysis allows the complex interrelated dynamics to be unravelled systematically. In particular, when the narrative gets messy because of the confluence of storylines, the MPA allows a methodical way to understand the interlinkages and interdependencies.

The classic problem of temporal delineation of a transition, in other words when does a transition begin and end, evaporates when using the MPA. A pattern, being specific change happening to constellations under certain conditions, is a more discernible unit of analysis than a transition. The methodology naturally suggests periods of transition—'pivot periods', those where the storylines converge—allowing the analyst to refine the temporal demarcation when necessary.

The MPA provides rigour to narrative analyses, to the point of allowing diagrammatic and formulaic representations of storylines. For example, the pattern schemas in Section 3 bridge the false divide between quantitative and qualitative approaches, enabling modelling exercises to be more directly comparable to narratives. The constellation and pattern concepts are easily parsed into a visual language like those in Figures 1,3 and 4. These are not only useful to computational approaches but also for intuitive communication of a complex story of system change.

While the MPA itself does not directly give policy guidance, insight from the analysis may suggest fruitful avenues in the pursuit of systemic change. Knowing that a certain pattern is ongoing in the system, a policy-maker may design approaches and instruments to reinforce desirable patterns and respond to current drivers—that is, not to resist an ongoing pattern, but rather to manage it in a controlled way, like suggested by Van Raak [25].

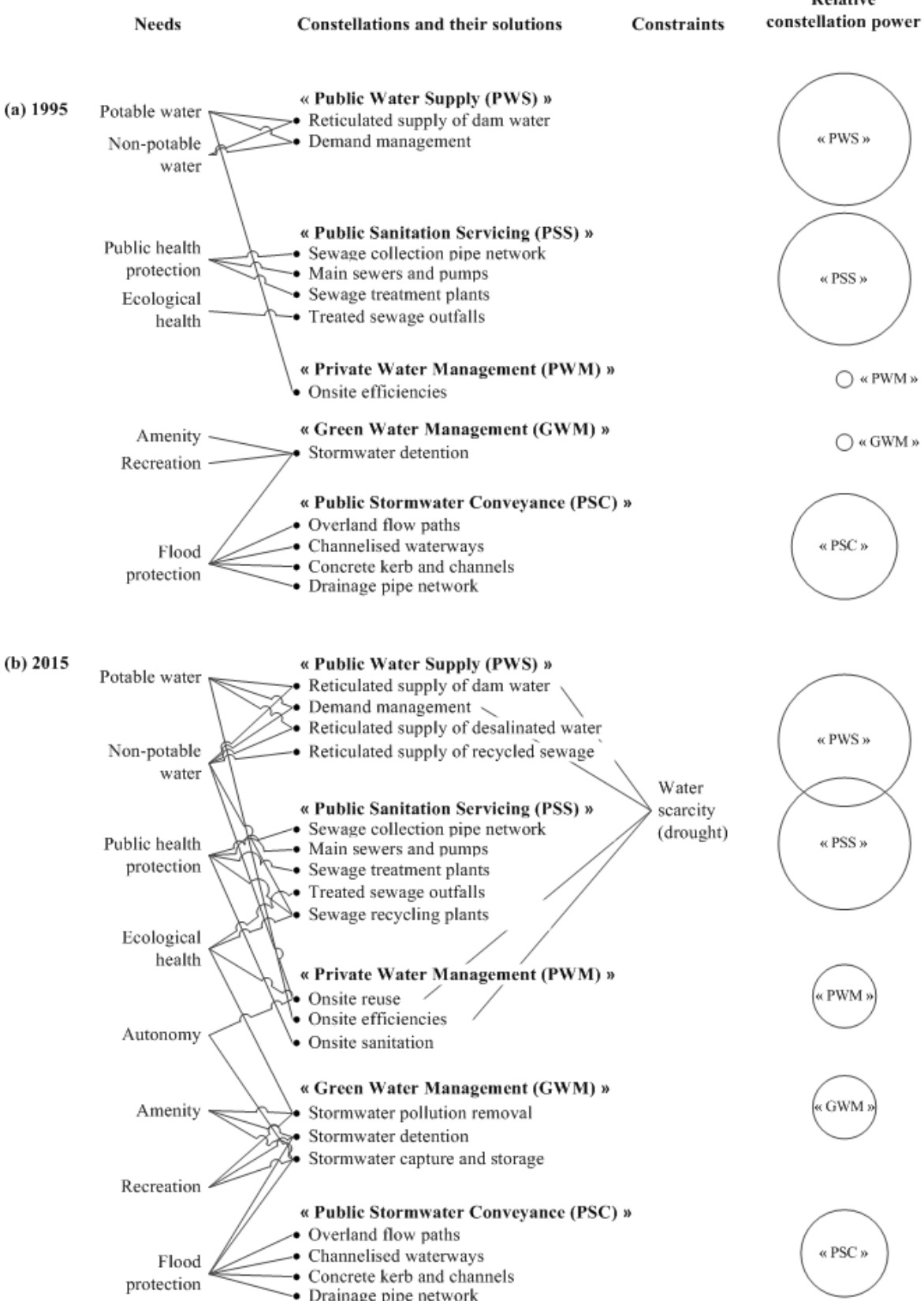

**Figure 4.** System configuration of Melbourne's water system in (**a**) 1995 and (**b**) 2015 to support detailed MPA analysis. A line between a need and a solution indicates that the solution meets that need. Similarly, for the constraint and solutions. The sizes of circles around the constellation names are indicative of their relative power.

**Author Contributions:** Both authors contributed equally to the conceptualisation and writing of this article.

**Funding:** This research received no external funding.

**Conflicts of Interest:** The authors declare no conflict of interest.

## Abbreviations

The following abbreviation is used in this manuscript:

MPA     Multi-Pattern Approach

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
