# Peer review of "The Multi-Pattern Approach for Systematic Analysis of Transition Pathways"

_sustainability, doi:10.3390/su11020318_

Reviewer 1 Report

This manuscript  is interesting and deserves to be published, but in the actual form it's too long and contain a certain quantity of unnecessary material within the main text.
In other word, in order to let the reader follow properly the text, I suggest to the Authors to move several tables and picture from the main text into an appendix. For example starting from pag. 16 (par 3.3.3 : Dynamics) onward, the Authors should only describe the methods and analyse the data, and fig.3 and following (e.g.: Analysis of Pathway 2 — Sanitation; Analysis of Pathway 3 — Water in the Environment, etc) should be moved into the Appendix.
Figure 4 is 4 pages long! the main text should be not more than 15-18 pages long, and all the unnecessary material should be canceled or put into an appendix. I believe that the Authors will easily rewrite this part of the article in a short time.

Author Response

Many thanks for your appreciation of our article. Regarding your comments about the volume of the manuscript, we agree that this is a lengthy manuscript. This is a direct consequence of the aim of the article, that is, to present a manual for a methodology and then to apply that manual in detail. We had checked in advance whether Sustainability set limits to manuscript length - which they do not. After correspondence with the editorial office we decided not to implement your suggestions.

Reviewer 2 Report

The paper presents the MPA methodology in order to provide a systematic analysis of transition

pathways with guidance on its application (first part of the paper) 

And an illustrative case study of Melbourne’s water system to proof the convenience of the proposal (second part)

The theoretical discussion of the model has been presented elsewhere by the authors but is claimed not to be the focus of the present paper. 

Overall, the paper is a well written paper on a novel approach to study transitions pathways, besides it is illustrated with a specific case study also done by the authors together with frequent illustrations of other studies. I congratulate the authors for the work.

The paper has, however, two weak points.

In the one hand (part one) the paper acknowledges other works on the topic (like the MLP by Geels) but ignores others like the functional approach by Bergek and other. It also fails to explain why this MPA that they propose is better than other models in the literature, and specially, it fails to acknowledge limitations of this proposal. Also, a graphical scheme of the MPA process (as in Bergek et al., Research Policy 37(2007), 407-429)

In the other hand (part two) the  illustrative case study of Melbourne’s water system lack a detailed mention to the methods and tools used for this part of the research (i.e. lines 394-407 does not show interview guide nor other tools used, neither a detailed account of secondary sources used).

Also footnote number 4 is redundant.  

Author Response

Many thanks for your review of our manuscript. You raise two main points. One is about a lack of discussion of e.g. the functional approach (TIS) by Bergek and others and why our proposal would be better. The other points out a lack of detailed mention of the empirical methods (e.g. interview guides) applied for the case study.

To start with the latter, the case study - though extensive and detailed - serves the purpose of illustrating the methodology and manual of the MPA. The empirical research itself was not the main message. We have however added a reference to a previous publication of ours on this case with a more detailed presentation of the empirical methods.

Regarding TIS and why our proposal would be better than other frameworks around, we think this may be a slight misunderstanding. The MPA is an approach to analyse transitions pathways. We do not think there are any extant frameworks for that purpose and therefore also do not think ours is better. There are certainly typologies of transitions and even typologies of transitions pathways. We discuss already discussed these in the manuscript. TIS is not a framework for the analysis of transitions pathways and it also does not proclaim to be. A comparative discussion of TIS would therefore be out of place we think.

Regarding Footnote 4, we think this etymological reminder of the true nature of analysis is relevant.